# National Pharmacist Workforce Study (NPWS): Description of 2019 Survey Methods and Assessment of Nonresponse Bias

**DOI:** 10.3390/pharmacy9010020

**Published:** 2021-01-13

**Authors:** Matthew J. Witry, Vibhuti Arya, Brianne K. Bakken, Caroline A. Gaither, David H. Kreling, David A. Mott, Jon C. Schommer, William R. Doucette

**Affiliations:** 1College of Pharmacy, University of Iowa, Iowa City, IA 52242, USA; matthew-witry@uiowa.edu; 2College of Pharmacy & Health Sciences, St. John’s University, Jamaica, NY 11439, USA; aryav@stjohns.edu; 3Pharmacy School, Medical College of Wisconsin, Milwaukee, WI 53226, USA; bbakken@mcw.edu; 4College of Pharmacy, University of Minnesota, Minneapolis, MN 55455, USA; cgaither@umn.edu (C.A.G.); schom010@umn.edu (J.C.S.); 5School of Pharmacy, University of Wisconsin, Madison, WI 53705, USA; david.kreling@wisc.edu (D.H.K.); david.mott@wisc.edu (D.A.M.)

**Keywords:** pharmacist, workforce, survey, nonresponse bias

## Abstract

National Pharmacist Workforce Studies (NPWS) have been conducted in the U.S. every five years since 2000. This article describes the online survey methods used for the latest NPWS conducted in 2019 and provides an assessment for nonresponse bias. Three waves of emails containing a link to the online survey were sent to a random sample of about 96,000 pharmacists licensed in the United States. The survey asked about pharmacist employment, work activities, work–life balance, practice characteristics, pharmacist demographics and training. A total of 5467 usable responses were received, for a usable response rate of 5.8%. To assess for nonresponse bias, respondent characteristics were compared to the population of U.S. pharmacists and a benchmark, while a wave analysis compared early and late respondents. The pharmacist sample–population comparison and the benchmark comparison showed that the NPWS respondents had a higher percentage of female pharmacists and a lower proportion of young pharmacists compared to the population of U.S. pharmacists and the benchmark sample. In some contrast, the wave analysis showed that the early respondents had a higher percentage of males and older pharmacists compared to the late respondents. Both the wave analysis and the benchmark comparison showed that the NPWS respondents (and early respondents) had a lower percent of pharmacists with a PharmD degree than did the late respondents and the benchmark group. These differences should be considered when interpreting the findings from the 2019 NPWS.

## 1. Introduction

The National Pharmacist Workforce Study (NPWS) has occurred in 2000, 2004, 2009, 2014, and most recently 2019 [1]. The study is overseen by the Pharmacy Workforce Center (PWC) which is comprised of U.S. national pharmacist and pharmacy associations. The mission of the PWC is “to serve the pharmacy profession and the public by actively researching, analyzing, and monitoring the size, demography and activities of the pharmacy workforce” [2]. Historically, the NPWS has provided data used to describe pharmacist labor supply [3,4,5,6,7,8], pharmacist work activities and work–life balance [9,10,11,12,13,14,15] and organizational influences on pharmacy practice and service delivery [16,17,18,19].

For the 2019 NPWS, there were two main changes from all previous NPWSs, which were mail surveys. First, this survey was delivered electronically to create survey distribution efficiencies and to allow tailoring of questions to specific subgroups of pharmacists based on demographics, practice settings and work status. Second, additional targeted sections were added to the core NPWS survey to address emergent topics such as pharmacist burnout, workplace harassment, naloxone dispensing and ambulatory care pharmacy practice. The survey also targeted questions specific to pharmacists who are retired and unemployed.

The purpose of this report is to describe the survey methods and respondent characteristics of the 2019 NPWS. Specific objectives were to (1) describe the 2019 National Pharmacists Workforce Survey protocol and (2) assess evidence about nonresponse bias for the 2019 NPWS.

## 2. Materials and Methods

For the 2019 NPWS, an electronic survey was preferred for two main reasons. First, an electronic survey has the potential to decrease the overall cost of the project because costs related to postage, processing and manual data entry are reduced [20]. Second, an electronic survey allows for respondents to receive questionnaire items tailored to their previous responses. For example, persons who respond they work as a hospital pharmacist would be presented with survey items relevant to hospital pharmacists but not to community-based pharmacists. The team considered these benefits against the challenges of electronic surveys, which generally have lower response rates [20,21]. The research team decided to use electronic surveying for the 2019 effort with the rationale that the benefits of question tailoring and efficiencies in survey administration would outweigh the limitations of a relatively low response rate.

Persons targeted for the 2019 NPWS were licensed U.S. pharmacists. The National Association of Boards of Pharmacy Foundation (NABPF) emailed a link to the electronic survey to a systematic random sample of 96,110, representing about 25% of all persons with an active U.S. pharmacist license. The subjects had to have an email listed in their record at the NABPF. Pharmacists licensed in multiple states were processed before the sampling to be listed only once in the sample frame.

The research team started with a core set of items from the 2014 NPWS, including pharmacist work characteristics, work environment, pharmacist work attitudes, work–life balance and future career plans. The team met via conference calls to develop the 2019 electronic survey. New topics included opioid-related questions, pharmacist burnout/fulfillment, discrimination and harassment in the pharmacy workplace and leadership in pharmacy. Subgroups created through branching and/or skipping questions included those based on practice setting (i.e., community, hospital), pharmacy managers/owners/administrators and retired pharmacists.

The questionnaire underwent usability testing by members of the research team and associates who followed different response paths to ensure readability and logical flow. Changes were made based on this testing. Next, the questionnaire was pilot tested on a sample of 2231 licensed pharmacists using a one-time email from the NABPF in a format like what would be used for the main survey. The response rate was assessed, as were items that appeared to be skipped or burdensome depending on where persons dropped out of taking the survey. Based on these findings, changes were made to the survey including condensing scales with seven response options to five response options. Also, some multi-item scales were revised, and some sections were rearranged so they occurred at different points in the survey flow.

For the main survey, all contacts occurred over the Internet. The NABPF sent three email contacts to the sample of 96,110 licensed pharmacists: an initial email contact and two reminder emails sent about two weeks apart. The survey emails briefly described the purpose and asked recipients to click on the survey link if willing to participate. Recipients of the email were told the survey was voluntary, anonymous, would take about 15–20 minutes depending on their characteristics and that they were free to skip items they did not want to answer. The email itself was not personalized and no monetary incentives were provided, although participants could enter their email into a separate survey to receive a synopsis of the survey findings. There was no additional promotion of the survey apart from persons in the sample receiving the email invitations. This study was approved by the University of Iowa Institutional Review Board.

The survey was administered using Qualtrics (Provo, UT, USA). Items in the survey were displayed in a uniform way without randomization. Conditional questioning for specific variable subsets was used. Most employed participants completing the survey answered over 100 items, depending on their characteristics. Pages contained between 1 and 8 items. Though most items did not require a response, it was required, however, that all employed respondents enter their work setting as that produced most of the conditional items. Most questions forced a single response and if multiple responses were allowed/desired, this was noted in the instructions for that item.

The following data were obtained from the NABPF for the 3 survey mailings that included survey links: the number of emails delivered, the number of emails opened, the number of emails where the survey link was opened and the number of undeliverable emails. As per previous NPWSs, a survey was considered usable if values were present for age, gender, practice setting, employment status, and hours worked per week. To assess for nonresponse bias, several analyses were conducted, including (1) comparison of sample and population, (2) wave analysis and (3) benchmarking [22,23].

For the comparison of sample to population, the characteristics of the group of usable responses (i.e., sample) were compared with those of the full random sample from the NABPF (i.e., population). For this assessment, comparisons were made between the respondents and the overall sample based on gender, geographic region and year of first licensure/pharmacy graduation. The respondents were asked the year of their first pharmacy license, while the NABPF supplied the year of pharmacy graduation. For the wave analysis, early respondents (first ten days of survey) were compared to late respondents (after the 3rd email) on age, gender, PharmD degree, employment status and year of first pharmacy license. The benchmarking was conducted by comparing the 2019 NPWS respondents to pharmacist respondents to the 2017 American Community Survey (ACS) on gender, age group, race and having a PharmD degree. The ACS data were taken from the U.S. Health Workforce Chartbook, which describes the demographic make-up of healthcare occupations in the U.S., including pharmacists [24]. All descriptive analyses were done using SPSS version 26 (IBM, Armonk, NY, USA).

## 3. Results

A total of 94,803 unique internet protocol (IP) addresses were verified to have received an email (Table 1). Of these, 8466 (8.95%) clicked on the link to open the survey. A total of 5467 usable responses were received for a usable response rate of 5.8%. Using the number of pharmacists who clicked on the survey link as a denominator, 64.6% provided a usable response set. The 2019 NPWS respondents had a higher percentage of female pharmacists, a greater percentage from the Midwest and had a greater percent graduated with a pharmacy degree by 2000 than the population (Table 2).

When comparing early to late respondents in the wave analysis, the early respondents were older, had a lower percentage of female pharmacists, had a lesser percentage with a PharmD degree and a lesser percentage practicing pharmacy (Table 3). Compared to the ACS respondents, the NPWS 2019 respondents had a higher percentage of female respondents, were older, had a greater percentage of white respondents and a lower proportion of respondents with a PharmD degree (Table 4).

## 4. Discussion

The 2019 NPWS survey response rate was lower than the mailed response rate from previous NPWSs, raising concerns about nonresponse bias. The online 2019 NPWS survey compared well with industry averages for online surveys of health professionals [24]. In addition, the relatively high rate of completed surveys by those who clicked on the survey link (64.6%) shows good interest in the survey topics. The use of an online survey resulted in the largest number of usable responses (N = 5467) in any of the five NPWSs conducted since 2000. This number of respondents will allow subgroup analyses of groups of pharmacists that have not been commonly analyzed in national samples due to sample size constraints, including ambulatory pharmacists, pharmacy owners and retired pharmacists.

The assessment of the nonresponse bias identifies some considerations about the generalizability of the data collected in the 2019 NPWS. All three of the comparisons showed that the NPWS respondents are older than the comparator groups (i.e., population of U.S. licensed pharmacists, late respondents, ACS respondents). That is, the data from the 2019 NPWS appear to underrepresent younger pharmacists. As a group, older pharmacists are likely to have more work experience, and would be expected to have less current debt from pharmacy school. It is difficult to determine how their work–life balance might differ from the somewhat younger population of licensed pharmacists. For example, older pharmacists who do not have children living at home might have less work–home conflict than younger parent pharmacists who need to manage both work and family childcare responsibilities. Also, older pharmacists typically have had longer since pharmacy school, and might not be up to date on the latest practice knowledge, which could affect their stress at work or the work tasks they perform.

Another result of the comparisons assessing for nonresponse bias is that the NPWS 2019 respondents had a higher percentage of female pharmacists than did the population of licensed pharmacists and the ACS sample. In contrast, the early wave of respondents had a lower proportion of female pharmacists than the late wave of respondents. This contrast in results illustrates a challenge in evaluating survey data for nonresponse bias. The comparison of the NPWS respondents with the population of licensed pharmacists provides the best view of the generalizability of the NPWS 2019 data. In this case, the 2019 NPWS data somewhat overrepresent female pharmacists.

A third variable showing differences between the NPWS respondents and the comparator groups was having a PharmD degree. The percentage of respondents in the early wave of respondents and the NPWS group was lower than in the late wave respondents and the ACS respondents. PharmD degree was not available in the descriptors of the population of licensed pharmacists. Given these results, the 2019 NPWS data underrepresent pharmacists with a PharmD degree, which is consistent with underrepresenting younger pharmacists.

This study has some limitations. The relatively lower response rate compared to mail administrations of previous NPWS raises concerns about nonresponse bias. However, we have described how the 2019 NPWS respondents differ from the population of licensed pharmacists. No actual survey of non-respondents to the full 2019 NPWS was conducted, which would have provided additional information for evaluating nonresponse bias.

## 5. Conclusions

The 2019 National Pharmacist Workforce Study collected over 5000 usable responses from licensed pharmacists in the U.S. While the response rate was low, the relatively large sample size will allow analyses of subgroups of pharmacists. The assessment of the nonresponse bias showed that the NPWS respondents had some differences compared to the population of U.S. pharmacists. These differences should be considered when interpreting the findings from the 2019 NPWS.

## Figures and Tables

**Table 1 pharmacy-09-00020-t001:** The characteristics of the three email waves sent to licensed pharmacists.

Date Sent	Total Recipients	Opened Email (%)	Clicked on Link (%)	Bounced (%)
22 May 2019	94,803	14,038 (14.8%)	2016 (14.4%)	2341 (2.5%)
31 May 2019	93,092	31,563 (33.9%)	3663 (11.6%)	850 (0.9%)
10 June 2019	92,845	31,014 (33.4%)	2787 (9.0%)	694 (0.75%)
NPWS Mean	93,580	27.3%	11.1%	1.4%
Healthcare Professional Mean ^1^	N/A	18.9%	4.5%	7.0%

^1^ Source: Constant Contact [24].

**Table 2 pharmacy-09-00020-t002:** A comparison of the NPWS respondent and population characteristics.

	Respondents n (%) *	Population n (%) *	Chi-square Test
**Gender**	**N = 5534**	**N = 96,110**	*p* < 0.01
Male	2098 (37.9)	39,975 (41.6)	
Female	3427 (61.9)	55,849 (58.1)	
Non-binary	9 (0.2)	NA	
Unknown	NA	286 (0.30)	
**Region of Country (Residence)**	N = 5342	N = 96,110	*p* < 0.01
Northeast	945 (17.7)	18,561 (19.3)	
Midwest	1298 (24.3)	21,205 (22.1)	
South	2008 (37.5)	36,997 (38.5)	
West	1066 (19.9)	18,818 (19.6)	
Outside the 50 U.S. & D.C.	25 (0.5)	529 (0.6)	
**Years of 1st Licensure or Pharmacy Graduation Date** **	**First Licensure** N = 5534	**Graduation Date** N = 94,322	*p* < 0.01
up to 1960	26 (0.5)	228 (0.2)	
1961–1970	156 (2.8)	1557 (1.7)	
1971–1980	733 (13.2)	7021 (7.4)	
1981–1990	1020 (18.4)	11,329 (12.0)	
1991–2000	940 (17.0)	15,909 (16.9)	
2001–2010	790 (14.3)	16,380 (17.4)	
2011–2019	1869 (33.8)	41,898 (44.4)	

* Percent figures are column percentages ** Note that first licensure could differ from pharmacy graduation date, which could create some differences in this comparison.

**Table 3 pharmacy-09-00020-t003:** Wave Analysis: A comparison of respondents of first e-mailing to respondents after the last e-mailing of survey.

	First E-Mail ^‡^ n (%) *	After Final E-Mail n (%) *	Chi-square Test
**Age**	N = 1223	N = 1932	*p* < 0.01
≤30	150 (12.3)	289 (15.0)	
31 to 40	237 (19.4)	559 (28.9)	
41 to 50	205 (16.8)	306 (15.8)	
51 to 60	294 (24.0)	400 (20.7)	
61 to 70	268 (21.9)	290 (15.0)	
> 70	69 (5.6)	88 (4.6)	
**Gender**	N = 1226	N = 1930	*p* < 0.01
Male	527 (43.0)	690 (35.8)	
Female	699 (57.0)	1240 (64.2)	
**Pharmd Degree**	N = 1226	N = 1930	*p* < 0.01
Yes	544 (44.4)	1062 (55.0)	
No	682 (55.6)	868 (45.0)	
**Employment Status**	N = 1226	N = 1930	*p* = 0.01
Practicing pharmacy	936 (76.3)	1541 (79.8)	
Healthcare—not practicing	63 (5.1)	92 (4.8)	
Non-Healthcare	12 (1.0)	10 (0.5)	
Retired	157 (12.8)	180 (9.3)	
Unemployed	58 (4.7)	107 (5.5)	
**Practice Setting**	N = 1009	N = 1759	*p* = 0.36
Community	454 (45.0)	837 (47.6)	
Outpatient/MD Clinic	59 (5.8)	103 (5.9)	
Hospital	271 (26.9)	438 (24.9)	
Other: patient care	99 (9.8)	192 (10.9)	
Other: not patient care	126 (12.5)	189 (10.7)	

^‡^ The first e-mail dates were 05/22/19-05/30/19 (9 days) and the third e-mail dates were 06/10/19–07/07/19 (28 days). * The percent figures reported are column percentages.

**Table 4 pharmacy-09-00020-t004:** Benchmarking: 2019 NPWS respondents compared to 2017 American Community Survey Respondents.

Characteristic	NPWS 2019 Respondents	ACS 2017 Respondents	Chi-square Test
**Gender (%)**			*p* < 0.01
Female	61.9	56.1	
Male	38.1	43.9	
**Age (%)**			*p* < 0.01
<30 years	15.4	19.4	
31–35	16.2	18.6	
36–40	9.6	11.7	
41–45	7.2	11.4	
46–50	9.4	10.0	
51–55	10.6	8.6	
56-60	10.3	8.3	
61-65	9.3	7.1	
66-70	7.0	3.4	
>70	5.0	1.5	
**Race (%)**			*p* < 0.01
White	78.2	71.1	
Asian	11.1	19.7	
Black	4.9	6.9	
Other	5.8	2.2	
**PharmD degree (% yes)**	65.4	72.1	*p* < 0.01

ACS results taken from the US Health Workforce Chartbook 2018 [25].

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
