# Peer review of "National Pharmacist Workforce Study (NPWS): Description of 2019 Survey Methods and Assessment of Nonresponse Bias"

_pharmacy, 2021, doi:10.3390/pharmacy9010020_

Round 1

Reviewer 1 Report

Thank you for providing this important context to the 2019 National Pharmacist Workforce Study (NPWS).  To better understand the overall results, it's important that readers understand the methods used to obtain the data and how representative the sample is of the general pharmacist population to which these results are being inferred.

Overall, the Introduction provides an appropriate level of background to understand how and why the study was done, and in particular the reason to make the transition from a mail administration to an online administration.

The methods also appropriately describe how the data was collected.  I do have a question regarding how the sample received the email and invitation to participate from the NABP Foundation.  My understanding is that in past administrations of the NPWS the survey was mailed to respondents from a university, with a cover letter from the researchers which made it clear that the purpose of the survey was to conduct pharmacy workforce research in an impartial manner.  In a quick internet search for "NABP Foundation", I could not find much about them other than that they are associated with the National Association of Boards of Pharmacy (NABP)...there is not a separate web site for the NABP Foundation which states their purpose.  Given that many respondents associate NABP with (1) their licensure exam, or (2) their state board of pharmacy, were there concerns about having the sample receive an email from the NABP Foundation asking for information about their work?  Was it clear to the sample that they were not providing data to a regulator or the the NABP Foundation?  Might having this request from the NABP Foundation, as opposed to university-based researchers, have inhibited the response rates, or the types of information respondents provided on the survey?

My only other concern is with Table 4, which makes comparisons between the sample and the ACS data for pharmacists.  Which statistical test was used to make this comparisons?  The other tables state that the comparisons were Chi-squared tests, but Table 4 does not indicate a test with the significance values.  In order to make a statistical comparison, wouldn't the researchers need to compare the 2019 NPWS data to the actual ACS dataset?

Author Response

Thank you for your comments. This work is intended to raise the issue of more consistently examining survey non-response bias when conducting nationally representative workforce research. Your feedback has helped us with that.

Overall, the Introduction provides an appropriate level of background to understand how and why the study was done, and in particular the reason to make the transition from a mail administration to an online administration.

The methods also appropriately describe how the data was collected.  I do have a question regarding how the sample received the email and invitation to participate from the NABP Foundation.  My understanding is that in past administrations of the NPWS the survey was mailed to respondents from a university, with a cover letter from the researchers which made it clear that the purpose of the survey was to conduct pharmacy workforce research in an impartial manner.  In a quick internet search for "NABP Foundation", I could not find much about them other than that they are associated with the National Association of Boards of Pharmacy (NABP)...there is not a separate web site for the NABP Foundation which states their purpose.  Given that many respondents associate NABP with (1) their licensure exam, or (2) their state board of pharmacy, were there concerns about having the sample receive an email from the NABP Foundation asking for information about their work?  Was it clear to the sample that they were not providing data to a regulator or the NABP Foundation?  Might having this request from the NABP Foundation, as opposed to university-based researchers, have inhibited the response rates, or the types of information respondents provided on the survey?

RESPONSE: The cover email stated that NABP Foundation was assisting with sending out the emails, but it was clear that the study was sponsored by the Pharmacy Workforce Center (PWC). PWC is an independent coalition of national pharmacy associations. The cover email stated the survey was anonymous and voluntary, and it was signed by the principal investigator using his university affiliation. It is possible that some subjects were concerned about NABP seeing their responses, but the emails tried to limit those concerns. The online survey was hosted at the University of Iowa, so NABP did not get any survey data.

My only other concern is with Table 4, which makes comparisons between the sample and the ACS data for pharmacists.  Which statistical test was used to make this comparisons?  The other tables state that the comparisons were Chi-squared tests, but Table 4 does not indicate a test with the significance values.  In order to make a statistical comparison, wouldn't the researchers need to compare the 2019 NPWS data to the actual ACS dataset?

RESPONSE: We manually calculated the chi-squares for the comparisons in Table 4. We have edited Table 4 to show that a chi-square test was used for the comparisons.

Reviewer 2 Report

Thank you for having the opportunity to review the manuscript "National Pharmacist Workforce Study (NPWS): 3 Description of 2019 Survey Methods and Assessment 4 of Nonresponse Bias". 

The period taken into consideration is a strong point of the manuscript so, in order to be suitable for publication, I recommend to the authors to take into consideration the following comments:

  • abstract must contain some important results - only mentioning the distribution among genders s not enough. In pharmacy field - there are more women than men in all countries - so this result is not a new one.
  • the abstract is larger than the Introduction section .....please do a proper Introduction.
  • the purpose if the study is  " The purpose of this report is to describe the survey methods and respondent characteristics of
    46 the 2019 NPWS. Specific objectives were to 1) describe the 2019 National Pharmacists Workforce Survey protocol and 2) assess evidence about nonresponse bias for the 2019 NPWS" ....... in this case please revise the title of the manuscript.
  • In Material and Methods section is a large amount of paragraphs that present results and suppositions. Please move them into Results section and into Discussion section (lines 59-63, ,...)
  • Results section is really poor in research results. Only socio-demographic data....with no comparative analysis or other statistical important analysis in order to generate important/new results. Please try to extract more information. Socio-demographic data represent not so "Important" results.
  • please revise Conclusion section - results must be eliminated. Conclusion section must refer to authors' interpretation of data and a general idea"a as a results and recommendation.

Author Response

Thankyou for your comments. This work is intended to raise the issue of more consistently examining survey non-response bias when conducting nationally representative workforce research. Your feedback has helped us with that.

Reviewer 2

Thank you for having the opportunity to review the manuscript "National Pharmacist Workforce Study (NPWS): 3 Description of 2019 Survey Methods and Assessment 4 of Nonresponse Bias". 

The period taken into consideration is a strong point of the manuscript so, in order to be suitable for publication, I recommend to the authors to take into consideration the following comments:

  • abstract must contain some important results - only mentioning the distribution among genders s not enough. In pharmacy field - there are more women than men in all countries - so this result is not a new one.

RESPONSE: Comparison results about gender, age and PharmD degree are now in the abstract.

  • the abstract is larger than the Introduction section .....please do a proper Introduction.

RESPONSE: This manuscript is a Brief Report, which necessarily limits the length of some sections. The purpose of this Introduction was to briefly describe the contributions made by previous National Pharmacist Workforce Studies and point to the need/desire to change from a mail survey to an electronic one. While we could expand on either of these, we think the current Introduction accomplishes our purpose while staying within reasonable limits for the Brief Report. Thus, we have not lengthened the Introduction.

  • the purpose if the study is  " The purpose of this report is to describe the survey methods and respondent characteristics of
    46 the 2019 NPWS. Specific objectives were to 1) describe the 2019 National Pharmacists Workforce Survey protocol and 2) assess evidence about nonresponse bias for the 2019 NPWS" ....... in this case please revise the title of the manuscript.

RESPONSE: The current title closely addresses the specific objectives. Thus, we have not changed the manuscript title.

  • In Material and Methods section is a large amount of paragraphs that present results and suppositions. Please move them into Results section and into Discussion section (lines 59-63, ,...)

RESPONSE: There are not results in the Materials and Methods section. It is unclear to what this comment is referring. The lines noted (59-63) are included in this section to support the use of the electronic survey mode for the 2019 NPWS.

  • Results section is really poor in research results. Only socio-demographic data....with no comparative analysis or other statistical important analysis in order to generate important/new results. Please try to extract more information. Socio-demographic data represent not so "Important" results.

RESPONSE: These Results help us meet the objectives of this paper. We consider the data reported to be important since they show evidence regarding potential non-response bias for the 2019 NPWS findings.

  • please revise Conclusion section - results must be eliminated. Conclusion section must refer to authors' interpretation of data and a general idea"a as a results and recommendation.

RESPONSE: We have removed from the Conclusion the specific differences identified in assessing for evidence of potential non-response bias.

Reviewer 3 Report

Thank you for the opportunity to review “National Pharmacist Workforce Study (NPWS): Description of 2019 Survey Methods and Assessment of Nonresponse Bias”. This was a well-written and concise piece of work that described nonresponse bias assessments of the 2019 NPWS. Just a few suggestions for clarification have been included, below.

ABSTRACT

  1. In the abstract, it isn’t clear as to why the authors chose to conduct a nonresponse bias assessment on the 2019 NPWS, or why that would be important to report as its own article. However, that became clearer to me in reading the introduction as the 2019 revised methods were described. I know there are word count issues at hand in the abstract, but the second sentence of the abstract could read, “This article describes the [revised] methods used for the latest NPWS conducted in 2019…” Maybe you could save a word by removing “the” out of line 25.

  1. The abstract appears to be describing 2 methods for assessing non-response bias while the methods section is describing 3 methods. So, when the abstract is describing “two benchmarks” in line 21, is that to imply both the benchmark analysis and the comparison of sample and population described in line 102? Can this be clarified?

  1. The last sentence of the abstract seems a little too specific for all the results reported in this paper. I think if it was more like the last sentence of the conclusions section, that would be more appropriate. It might also save a few words to clarify the comment listed above.

  1. The ampersand (&) in line 19 should be written as “and”.

INTRODUCTION

  1. In line 55, downsides may be better written as “challenges”.

RESULTS

  1. In line 121, describing the results of Table 2, it is noted that the 2019 NPWS respondents were older than the population. I don’t know that this is totally appropriate to describe a variable that was looking at year of first licensure or graduation date. That would be an assumption that they are older, but probably should not be described that way in the text without indicating the caveat that you are making that assumption.

Author Response

Thankyou for your comments. This work is intended to raise the issue of more consistently examining survey non-response bias when conducting nationally representative workforce research. Your feedback has helped us with that.

Reviewer 3

Thank you for the opportunity to review “National Pharmacist Workforce Study (NPWS): Description of 2019 Survey Methods and Assessment of Nonresponse Bias”. This was a well-written and concise piece of work that described nonresponse bias assessments of the 2019 NPWS. Just a few suggestions for clarification have been included, below.

ABSTRACT

  1. In the abstract, it isn’t clear as to why the authors chose to conduct a nonresponse bias assessment on the 2019 NPWS, or why that would be important to report as its own article. However, that became clearer to me in reading the introduction as the 2019 revised methods were described. I know there are word count issues at hand in the abstract, but the second sentence of the abstract could read, “This article describes the [revised] methods used for the latest NPWS conducted in 2019…” Maybe you could save a word by removing “the” out of line 25.

 RESPONSE: We have edited the 2nd sentence in the Abstract to read, “This article describes the online survey methods used for the latest NPWS conducted in 2019 and provides an assessment for nonresponse bias.”

  1. The abstract appears to be describing 2 methods for assessing non-response bias while the methods section is describing 3 methods. So, when the abstract is describing “two benchmarks” in line 21, is that to imply both the benchmark analysis and the comparison of sample and population described in line 102? Can this be clarified?

 RESPONSE: We have edited the Abstract to clarify that three comparisons were made when assessing for potential nonresponse bias.

  1. The last sentence of the abstract seems a little too specific for all the results reported in this paper. I think if it was more like the last sentence of the conclusions section, that would be more appropriate. It might also save a few words to clarify the comment listed above.

 RESPONSE: We have changed the last sentence of the Abstract to be more generalized like the last sentence in the Conclusion.

  1. The ampersand (&) in line 19 should be written as “and”.

RESPONSE: We have made that change.

INTRODUCTION

In line 55, downsides may be better written as “challenges”.

RESPONSE: We have made that change.

 RESULTS

 In line 121, describing the results of Table 2, it is noted that the 2019 NPWS respondents were older than the population. I don’t know that this is totally appropriate to describe a variable that was looking at year of first licensure or graduation date. That would be an assumption that they are older, but probably should not be described that way in the text without indicating the caveat that you are making that assumption.

RESPONSE: We have edited that sentence [lines 123-125] to remove age and insert, “a greater percent graduated with a pharmacy degree by 2000”.

Round 2

Reviewer 2 Report

I found no important changes, especially the results are not interesting for the readers. None of the recommendations was included.

I consider that the manuscript is well written but the results are so poor. Maybe the presentation of the results would be more interesting for a Conference Presentation, but for a brief report published in a scientific journal.

I appreciate that is not enough (age, gender, year of licence, pharmacy degree, employment status and practice settings) - there are 6 socio-demographic characteristics that are presented in the brief report.

That is why I recommend that the final decision belongs to the Academic editor